# Healing of Unilateral Maxillary Sinusitis by Endodontic and Periodontal Treatment of Maxillary Teeth

**DOI:** 10.3390/medicina58091302

**Published:** 2022-09-18

**Authors:** Klaudia Migas, Remigiusz Kozłowski, Aleksandra Sierocka, Michał Marczak

**Affiliations:** 1Department of Management and Logistics in Healthcare, Medical University of Lodz, 90-419 Lodz, Poland; adreslewska@wp.pl (A.S.); michal.j.marczak@gmail.com (M.M.); 2Center of Security Technologies in Logistics, Faculty of Management, University of Lodz, 90-237 Lodz, Poland; remigiusz.kozlowski@umed.lodz.pl

**Keywords:** endodontics, periodontics, maxillary sinus, sinusitis, maxilla

## Abstract

Inflammatory conditions of dental origin may spread to the bone tissue, causing its destruction, and to anatomical structures located in the vicinity of the tooth affected with inflammation. Maxillary premolars and molars may develop inflammatory lesions of the Schneiderian membrane and lead to tooth-borne lesions in the maxillary sinuses. Unilateral inflammation of the maxillary sinuses should be diagnosed and treated. The aim of this study was to determine whether and after what time from the applied endodontic or nonsurgical periodontal treatment the inflammation in the maxillary sinus was diminished (assessed by the decrease in the Schneiderian membrane hypertrophy). A retrospective study was performed to analyze the records of endodontically, periodontally, or endodontically-periodontally treated patients with unilateral inflammation of the maxillary sinuses along with diagnostic Cone Beam Computed Tomography. The method for determination of the inflammation was measurements registered in millimeters in Carestream software. The analysis included the situation before treatment and 3 months, 6 months, and 12 months after completion of the treatment. Regardless of the origin of the maxillary sinus lesion, healing of inflammation of the sinus has been reported after the implementation of causative treatment of the maxillary tooth. Dental treatment reduces the need to implement conservative or surgical ENT treatment.

## 1. Introduction

Maxillary sinusitis is one of the most common causes of pain that patients usually associate with lesions of the upper premolars and molars. The most commonly reported complaints include pain in the area of one or more teeth with roots located in the vicinity of the inflamed maxillary sinus. The pain is moderate, sometimes escalating to severe, with tenderness to palpation and flares to cold. Additionally, when the maxillary sinuses are inflamed, there is palpation tenderness of the anterior wall of the maxillary sinus, headaches, and increased discomfort with anterior tilt [1].

Marginal and periapical periodontitis can cause infection of the maxillary sinus mucosa. In addition, the emerging blockage of the middle nasal meatus, which is usually caused by reactive mucosal proliferation, can lead to the retention of mucosal contents in the maxillary sinus, which in turn, can lead to bacterial infection. The restricted flow of sinus secretions promotes chronic inflammation. Symptoms occurring in acute maxillary sinusitis compared to those in the chronic phase are similar but exhibit significant severity without any periods of remission or exacerbation. Characteristic symptoms, apart from tenderness on palpation and aggravation of discomfort while changing the position of the head, may include discharge from the nasal cavity on the inflamed side, swelling, redness of the skin, fever, a feeling of weakness, and nausea. Lack of treatment of maxillary sinusitis may result in worsening of the infection or its penetration to the other paranasal sinuses and the surrounding anatomical structures, e.g., orbit, skull base. It is associated with serious complications such as encephalitis, meningitis, cavernous sinus thrombosis, and optic neuritis [2].

Classification of the types of periapical lesions with respect to their size and the presence of associated periodontal lesions includes Classes A, B, and C, which refer to teeth with a current lesion of endodontic origin, and Classes D, E, and F, which refer to teeth with a lesion of endodontic origin associated with periodontal lesions [3]. It should be emphasized that periapical lesions are, many times, asymptomatic, and an X-ray examination is the only way to visualize the lesion [4]. If the radiographic lesion is not healed or is even larger after one year, it is unlikely to ever heal. If the lesion is still present after six months, but has already diminished, it may be a sign of healing, but a follow-up visit is still necessary. Radiological determination of treatment success is confirmed if radiolucent areas are eliminated a minimum of one year after treatment [5]. A weakness of assessment based on the radiographic image is that X-ray interpretation is often dependent on personal predisposition and changes in angulation, as well as differences in interpretation by observers. Hence, it is suggested to use the periapical index to evaluate radiographic findings [6,7].

The dental pulp is usually vital when periodontal disease coexists until the accessory canals come into contact with the oral cavity environment. In this case, microorganisms can enter the pulp through these canals and cause an inflammatory reaction that can lead to pulp necrosis [8,9]. Antiseptic root canal treatment involves the elimination of infected pulp and dentin from the tooth by means of chemo-mechanical treatment of the root canal system and subsequent tight obturation. The prognosis for endodontic treatment is good if performed according to the current standards [10]. Periodontal therapy is based on nonsurgical procedures including mechanotherapy, i.e., SRP (scaling and root planning), which is the foundation of causative periodontal treatment. Its goal is to minimize the impact of plaque on the periodontium and reduce inflammation. Successful nonsurgical periodontal treatment leaves only isolated regions of periodontal tissues that will require additional periodontal intervention [11,12].

The main role in the pathogenesis of tooth-borne sinusitis is played by bacteria found in the oral cavity, especially those that cause pulpitis. The type of bacterial flora influences the nature of maxillary sinusitis, which can present in either a serous or purulent form. In the case of chronic sinusitis, apart from the type of bacterial flora, the type of sinus lesion formed is influenced by the duration of inflammation, its severity, and the exacerbation of the inflammatory process. In these conditions, it is possible to observe a hyperplastic form or polyps, with the possible presence of serous and purulent discharge [13]. In addition to the biological resistance of the Schneiderian membrane, recurrent acute inflammation, obstruction of the natural orifices of the sinus complex, and decreased absorption of the produced exudate play a significant role in maintaining and perpetuating the chronic condition [14]. Diagnosis using Cone Beam Computed Tomography facilitates the detection of potential foci of infection in the tissues adjacent to the maxillary sinus [15,16]. Inflammation of the maxillary sinuses is associated with local and general symptoms in patients. When a patient presents to the dentist’s office with symptoms characteristic of sinusitis, sinus diagnostics should be implemented. Symptoms accompanying inflammation of the maxillary and paranasal sinuses are usually associated with poor drainage and ventilation [17,18].

Patient complaints in cases of acute maxillary sinusitis include pain in the projection of an inflamed sinus, decreased mood, respiratory distress, fever, nasal discharge, and soreness to palpation of the inflamed sinus. In chronic inflammation, these complaints may be negligible or absent [13,17]. The diagnosis is established using anamnesis, physical examination, and radiological images. X-ray diagnostics of the maxillary sinuses demonstrate the degree of sinus involvement, including hypertrophy of the Schneiderian membrane or the presence of fluid in the sinuses. If Schneiderian membrane hypertrophy is confirmed, the condition may be treated endodontically and/or periodontally by the dentist. After periodic radiographic follow-ups, the hypertrophy may or may not be referred for ENT treatment, depending on the healing process. Treatment of inflammation of the maxillary sinuses begins with standard treatment, i.e., tooth extraction or endodontic treatment. Chronic Schneiderian membrane hypertrophy may require surgical treatment by the ENT specialist. ENT surgical treatment consists of gaining access to the maxillary sinus, either through the maxillary alveolar process or endoscopically through the anterior nostrils, removing the inflamed Schneiderian membrane and the contents of the inflamed maxillary sinus, and widening the outflow to the anterior nostrils in the medial wall of the maxillary sinus. Collaboration between the dentist and the ENT specialist is important for the successful treatment of tooth-borne inflammation of the maxillary sinuses [10,19,20,21].

The prevalence of tooth-borne sinusitis is 25.1% for pulp canal infections and 8.3% for marginal periodontitis [22].

The aim of this study was to determine the time after which Schneiderian membrane hypertrophy decreased after endodontic treatment and periodontal treatment. The hypothesis was that there is a possibility of X-ray diagnosis earlier than six months after treatment. 

## 2. Materials and Methods

This study employed a retrospective analysis of medical records of patients treated between January 2019 and January 2022. The study was carried out in accordance with the STROBE guidelines. Due to the nature of the study and the fact that medical data were analyzed without including the patient directly in the study, ethics committee agreement was not necessary. It is by virtue of national legislation: Art. 24. Act on patient’s rights and the Patient’s Rights Ombudsman, Art. 3. The Act of 15 April 2011 on medical activity, Art. 30. The Act of 29 June 1995 on Public Statistics.

The study enrolled the medical records obtained from 474 patients. Patients who were enrolled in the study had Cone Beam Computed Tomography (CBCT) scans of the maxilla before treatment and repeated at 3 months, 6 months, and 12 months after treatment. Each of the qualified patients had unilateral inflammation of the maxillary sinus prior to treatment, demonstrated by the CBCT scan of the maxilla. 

The inclusion criteria were as follows:Patients were in the age range from 18 to 65 years.Upper left or right premolar or molar tooth with indications to be primary or secondary root canal treatment.Upper left or right premolar or molar tooth with or without periodontal problems.Upper left or right premolar or molar tooth with inflammation of the maxillary sinus in the projection of the root tip of the treated tooth.Primary or secondary root canal treatment obturated using epoxy resin as a sealant.Root canal treatment performed in a single visit.

The exclusion criteria were as follows:

Drug use for any systemic disease and maxillary sinusitis in the last twelve months.Systematic disease and sinusitis in the last twelve months.Allergy in the last twelve months.Patients with a history of sinusitis.Patients with a bilateral maxillary sinusitis.Patients with ENT treatment in the medical history.Patients with dental implants in the treated side of maxilla.Patients below 18 years old and over 65 years old.Root canal treatment performed in a multiple-visit protocol.

Root-canal-treated teeth on multiple visits were excluded from the study in order to eliminate the additional factor and its impact, which could be reinfection of the root system between visits. During multi-session root canal treatment, there is a risk of infection of the root system through a leaky dental filling or falling out.

The sample size calculation was performed. The confidence level for the study was 95%, the margin error was 5%, the population proportion was 16%, and the population size was 4275. This means 198 or more measurements/surveys are needed to obtain a confidence level of 95% that the real value is within ±5% of the measured/surveyed value.

In the study group, primary endodontic treatment and retreatment were performed in a single visit. In the case of a tooth with a periodontal problem, nonsurgical periodontal treatment in the form of manual curettage and ultrasonic SRP was performed in a single visit before endodontic treatment. Periodontal status was assessed based on periodontal examination at 6 points around the tooth, the degree of tooth mobility, the probing depth, bleeding on probing, and the degree of gingival recession. Measurements were conducted with a manual periodontal probe calibrated in 1 mm increments.

Cone beam tomography was performed on a Carestream 9300 Premium C device (Carestream Dental LLS, Atlanta, GA, USA) and measurements were registered in millimeters in Carestream software (Carestream Dental LLS, Atlanta, GA, USA). The measurements were registered with a resolution of 180 microns. To evaluate the status of the maxillary sinus, a value of more than 2 mm of Schneiderian membrane thickening was interpreted as the presence of inflammation in the maxillary sinus. Inflammation of the maxillary sinus was assessed in the projection of the root tip of the treated tooth from the lowest point to the highest point at two measurement points. The highest point of the maxillary sinus mucosal proliferation was considered the highest point and the floor of the inflamed maxillary sinus was considered the lowest point.

Teeth and periapical tissues were assessed in the axial, coronal, and sagittal CBCT planes. Mucosal thickening was evaluated above the treated tooth in the axial, coronal, and sagittal CBCT planes. The thickening of the maxillary sinus mucosa was analyzed at the highest thickness from the maxillary sinus floor in the projection of the treated tooth (Figure 1).

All measurements were performed according to a standardized method [16]. For each CBCT study, two measurements were taken twice at a monthly interval. The CBCT scan was evaluated by two independent dentists, including a general dentist and a specialist in endodontics. When the difference between the observers was >0.2 mm, a new measurement was performed, and the average was determined based on this measurement. That was later included in the statistical analysis. The intraclass correlation coefficient (ICC) was measured for the dentists who assessed CBCT and was used to assess the reproducibility of the measurement. When ICC was >0.80, it was considered a satisfactory score. Reassessment was conducted when ICC was lower. The assessment was performed in a blinded manner. Before the CBCT assessment was performed, basic monitor tests were carried out: Threshold contrast of the displayed image, geometric correctness of the image, image viewing conditions, and image quality. For the CBCT device, the tests carried out included artifacts, the HU value, image homogeneity, noise level, high contrast resolution, and geometric correctness of the image. Dentists who evaluated the CBCT images had completed training in the assessment of CBCT images.

The calculations were performed using Statistica 12 (StatSoft, Cracow, Poland) and StatXact (Cytel, Cambridge, Massachusetts, USA) software. The Shapiro–Wilk test was used to check the distribution of the variables. The homogeneity of variance was checked with the Levene test. For comparisons between groups, the unpaired Student’s *t*-test was used. For in-group comparisons, a paired Student’s *t*-test was used. Furthermore, the effect size of Cohen’s D was calculated to indicate the strength of the relationships between the results. The results were considered statistically significant if *p* < 0.05. 

## 3. Results

### 3.1. Characteristics of the Study Group

Among the eligible medical records of 474 patients, 59% (280 medical records) were obtained from women and 41% (194 medical records) were from men. The patients whose records were analyzed were aged between 18 and 65 years (Table 1). There were 61% of medical records from primary root canal treatment and the rest were from retreatment. Based on the study group, there were more patients with treated molars than premolars and periodontal problems during the primary root canal treatment. However, during the retreatment, there were no such disproportions (Table 1). There were also no statistically significant differences between males and females in the cases of treated types of tooth or frequency in the occurrence of periodontal problems (Table 1).

Among the patients with periodontal problem, there were three grades of teeth mobility observed, according to the Mobility Index by Graces and Smales. Grade 0 mobility was observed in 146 of the analyzed teeth with a periodontal problem (55% of the cases), grade I mobility was seen in 108 teeth (42% of cases), and grade II mobility was seen in 8 teeth (3% of cases). Teeth with grade III mobility were not observed in the study.

### 3.2. Analysis of Time after Treatment for Decrease in the Schneiderian Membrane Hypertrophy

The time period in which there was a statistically significant reduction in the inflammation in the maxillary sinus since the start of treatment depended on the baseline condition of the tooth and surrounding tissues and the type of treatment provided. The highest value of the Schneiderian membrane recorded on the day of treatment was 28.3 mm and the lowest value was 5.2 mm. The greatest reduction in the Schneiderian membrane hypertrophy in the whole study group was observed after 12 months (*p* < 0.001; d = 1.92) (Figure 2). 

Before treatment, the Schneiderian membrane was 12.1 ± 4.5 mm and 4.9 ± 2.9 mm after 12 months. In patients after primary root canal treatment, the greatest reduction in inflammation for premolars and molars was observed after 12 months of observation for both patients with and without periodontal problems (Figure 3). In patients with periodontal problems, the thickness of the membrane was 12.2 ± 4.5 mm before treatment and 5.4 ± 3.3 mm after 12 months (*p* < 0.001; d = 1.75) (Figure 3). In patients without periodontal problems, the thickness of the membrane was 11.5 ± 4.2 mm before treatment and 4.9 ± 3.1 mm after 12 months with (*p* < 0.001; d = 1.81) (Figure 3).

In the group of premolars and molars affected by periodontal problems undergoing primary root canal treatment, there was a statistically significant reduction in unilateral inflammation of the maxillary sinus on the treatment side during the first postoperative period, i.e., from the end of treatment to 3 months (*p* < 0.001; d = 0.79). No statistically significant reduction in sinus inflammation was observed in this group of treated teeth from 3 months to 6 months. However, there was a statistically significant reduction in sinus inflammation from 6 months to 12 months (*p* < 0.001; d = 0.82). For maxillary premolars and molars without periodontal problems treated by primary root canal therapy, there was a statistically significant reduction in inflammation in the maxillary sinus on the treatment side between 0 and 3 months (*p* < 0.001; d = 0.72) and 12 months after the completion of treatment (*p* < 0.001; d = 0.95). In the same group, there was no statistically significant reduction in inflammation in the maxillary sinus between 3 and 6 months after completion of the primary root canal treatment (Figure 3).

There was no statistically significant difference between the thickness of the membrane at baseline values and values after 12 months in patients with and without periodontal problems during the primary root canal treatment (Figure 3).

In patients receiving endodontic retreatment, the greatest reduction in inflammation for premolars and molars was observed after 12 months of observation for both patients with and without periodontal problems (Figure 4). In patients with periodontal problems, the thickness of the membrane was 12.4 ± 4.3 mm at baseline and 4.2 ± 2.4 mm after 12 months (*p* < 0.001; d = 2.36). In patients without periodontal problems, the thickness of the membrane was 12.8 ± 4.9 mm at baseline and 4.5 ± 2.7 mm after 12 months (*p* < 0.001; d = 2.09) (Figure 4).

Endodontic retreatment of maxillary premolars and molars with periodontal lesions resulted in a statistically significant reduction in the level of unilateral inflammation in the maxillary sinus on the treatment side in all three periods: 0 to 3 months (*p* < 0.001; d = 0.97), 3 to 6 months (*p* < 0.01; d = 0.53), and 6 to 12 months after treatment (*p* < 0.001; d = 1.12)). For endodontic retreatment of maxillary premolars and molars without periodontal problems, a statistically significant reduction in unilateral sinusitis on the treated side occurred between the end of treatment and 3 months (*p* < 0.001; d = 0.86), between 3 and 6 months (*p* < 0.01; d = 0.55), and between 6 and 12 months after treatment (*p* < 0.001; d = 0.99) (Figure 4).

There was no statistically significant difference between the thickness of the membrane at baseline values and values after 12 months in patients with and without periodontal problems during the primary treatment (Figure 4).

## 4. Discussion

The goal of modern treatment is to minimize the invasiveness and maximize the effectiveness and safety of procedures performed for the patient, supplemented by reliable information on treatment options and their prognosis, so that the patient can make an informed treatment decision [23,24]. Collaboration between specialists from different medical and dental disciplines is now required to ensure optimal therapy and treatment outcomes [25,26,27]. The treatment of tooth-borne inflammation of the maxillary sinuses is complex and comprehensive [28].

Diagnosis of tooth-borne inflammation and controlling the healing process can be difficult, especially in the molar region, due to the overlap of anatomical structures. In the case of doubts while analyzing the 2D X-ray, it seems reasonable to implement diagnostic CBCT [29,30,31,32]. Successful diagnosis of unilateral tooth-borne sinusitis requires a dental examination in addition to an X-ray diagnosis [33].

In the case of suspicion of tooth-borne sinusitis that does not heal after conservative treatment, the ENT specialist should refer the patient to a dentist and work closely together to treat the inflammation [34]. This is all the more important because it may be difficult to make the diagnosis for the ENT physician alone [35], and failure to implement dental treatment for tooth-borne sinusitis may imply a lack of effective treatment [22]. Tooth-borne inflammation of the maxillary sinus must be specifically considered in cases of unilateral maxillary sinusitis [36]. Modern microscopic endodontics and periodontics allow for effective causative treatment. Properly performed root canal therapy and nonsurgical periodontal treatment help to reduce the unilateral inflammation of the maxillary sinus on the treatment side [37]. Dental treatment that eliminates the cause of inflammation is effective in treating inflammation of the maxillary sinuses [38].

Different research confirms or not that the presence of periodontal disease may be correlated with chronic maxillary sinusitis and that treatment of periodontal disease may be associated with a beneficial effect on maxillary sinus status [13,39,40]. In the present study, there was a slightly higher prevalence of periodontal disease around the treated teeth (55%), compared to teeth without periodontal disease (45%). Other studies also confirm a higher incidence of maxillary sinusitis in the vicinity of teeth with periodontal problems [41,42].

There was no statistically significant difference in the reduction of unilateral inflammation in the maxillary sinus between the groups of teeth that received treatment with and without coexisting periodontal disease. In the present study, we noticed a higher proportion of teeth that required primary root canal treatment (61%) as compared to those requiring endodontic retreatment (39%). The lower proportion of premolars and molars that require endodontic retreatment in the area of unilateral inflammation of the maxillary sinus may be due to the fact that patients were more likely to choose to extract a tooth that was unsuccessfully treated with primary root canal therapy.

In the case of asymptomatic unilateral maxillary sinusitis, if there is a possibility of reducing the inflammation by non-invasive endodontic treatment instead of ENT surgery, it seems to be convenient for the patient to begin with dental treatment with clinical and X-ray follow-up to monitor the healing process. In the case of symptomatic maxillary sinusitis, it seems reasonable to start with ENT treatment, since performing endodontic treatment beforehand does not significantly benefit the patient [43].

The benefits for the patient who has asymptomatic unilateral sinusitis also include lower treatment costs when started with endodontic and periodontal treatment rather than ENT surgery. It reduces the rate of sinus surgery due to inflammation healing after endodontic-periodontal treatment [44]. Teeth that were more frequently associated with unilateral sinusitis were maxillary molars (52%), which is consistent with other studies [45,46,47].

According to the guidelines of the European Society of Endodontology, the recommended X-ray follow-up after endodontic treatment is 6 months after the completion of the treatment. When evaluating the healing of unilateral inflammation of the maxillary sinus, it seems reasonable to implement control radiographs to evaluate the healing process at another time. In the case of teeth with periodontal problems treated initially with root canal therapy, it seems reasonable to perform an examination 3 months after the end of the treatment. In the case of endodontic retreatment of teeth with no periodontal problems, it seems optimal to perform control CBCT of the treated area after 3 months. In the case of endodontic retreatment of teeth with periodontal problems and teeth without periodontal problems subjected to primary root canal therapy, a period of 12 months after the completion of treatment seems to be the optimal time for tomographic control of the maxillary sinus on the treatment side.

This study is unique in terms of evaluating the healing process in the maxillary sinus after dental treatment at different periods after the completion of this treatment. Investigating the time needed for healing occurred in the maxillary sinus is important for dentists and ENT specialists to estimate how long the reasonable waiting period before possible surgical ENT treatment is and whether the healing process is proceeding properly or the prognosis is rather questionable and surgical ENT treatment should already be implemented. Moreover, the ability to predict which cases can be diagnosed with CBCT earlier than the recommended 6 months is crucial. For cases of endodontic retreatment of teeth without periodontal problems and primary root canal treatment of teeth with periodontal problems, it allows for diagnostics after 3 months since the end of the treatment. In the case of a lack of healing, ENT treatment should be implemented. However, in cases with a longer healing time for which the optimal time to perform CBCT diagnostics was 12 months, i.e., endodontic retreatment of teeth with periodontal problems and primary root canal treatment of teeth without periodontal problems, it is advisable to wait longer than the recommended 6 months before deciding on ENT treatment. Other studies also confirmed a reduction in inflammation of the maxillary sinus after 3 months [48].

The main limitation of the study was the sample size; however, it was dependent on the medical records of the patients that met the inclusion criteria of the study. Therefore, it would be recommended to conduct further studies including individuals with general diseases and taking permanent medications or to compare the results with untreated patients. Furthermore, further research with an extended duration of observation is recommended.

After 12 months from the end of treatment, a reduction in the thickness of the Schneiderian membrane to 2 mm was observed in 71 patients (15%). For 231 (49%) patients, the thickness of the Schneiderian membrane decreased below 5 mm after 12 months from the end of treatment. It means that the inflammation in the maxillary sinus was not fully healed after 12 months for all patients. In the absence of symptoms and ailments from the maxillary sinuses, it may be subject to further observation and control appointments. If ENT treatment is required, the earlier reduction of inflammation in the maxillary sinus as a result of dental treatment facilitates ENT treatment.

The results of the present study are important not only to the dentists and ENT specialists who cooperate together during the treatment process, but also to the patient, both in terms of the cost of treatment and its invasiveness associated with surgical ENT treatment. It should be concluded that in the case of non-healing unilateral inflammation of the maxillary sinus after a period of follow-up since dental treatment, the implementation of surgical ENT treatment is indicated and effective in eliminating inflammation in the maxillary sinus [49].

## 5. Conclusions

The results of the study showed that within 12 months of the endodontic or non-surgical periodontal treatment, the inflammation is significantly reduced regardless of gender or the type of dental treatment. It seems that there is a possibility of earlier X-ray diagnosis of the healing of maxillary sinus inflammation and the lack of need for ENT treatment, which can result in a cost reduction for the patient and the health care system, no need for hospitalization, and thus reduce the risk of nosocomial infection. The results of the study also indicate that the inflammation in the maxillary sinus was not fully healed in every case, which is why periodic check-ups are so important in order to initiate laryngological treatment in the event of ailments as early as possible. In the absence of general symptoms in a patient and the presence of inflammation in a maxillary sinus, dentists should at first implement causal treatment for dental causes, i.e., root canal treatment and periodontal treatment. The use of antibiotics and steroids is reduced in the absence of the need for conservative ENT treatment, thereby reducing the risk of developing resistant strains of bacteria. Further studies analyzing this issue among other patients, i.e., those with general diseases who take medications permanently, are recommended. The obtained results are clinically important because they show the possibility of earlier X-ray follow-up than the recommended 6 months in the case of the evaluation of the healing process of periapical lesions. Further research analyzing the same issue based on the more extensive database would be recommended so that the results can be generalized to a larger patient population. It is important that dentists do not prematurely refer patients for ENT treatment.

## Figures and Tables

**Figure 1 medicina-58-01302-f001:**
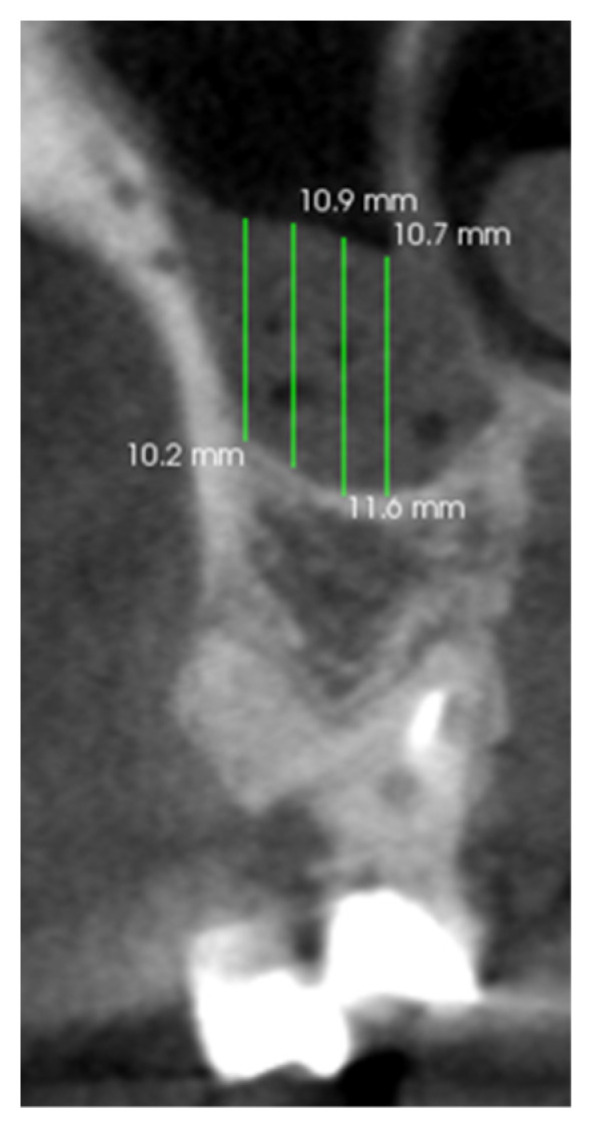
An image of a CBCT scan in coronal plane of the first maxillary molar and maxillary mucosal thickening measurement.

**Figure 2 medicina-58-01302-f002:**
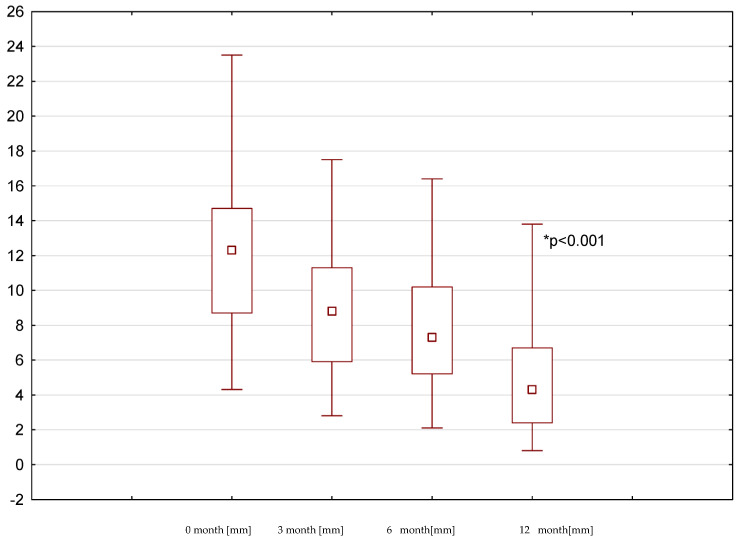
The thickness of the Schneider membrane in mm from 0 to 12 months of the treatment period.

**Figure 3 medicina-58-01302-f003:**
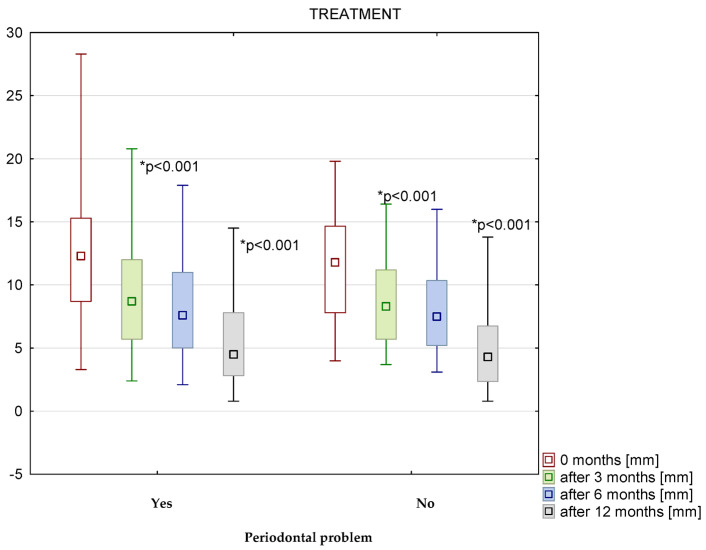
The thickness of Schneiderian membrane in mm from 0 to 12 months after primary treatment with and without periodontal problems.

**Figure 4 medicina-58-01302-f004:**
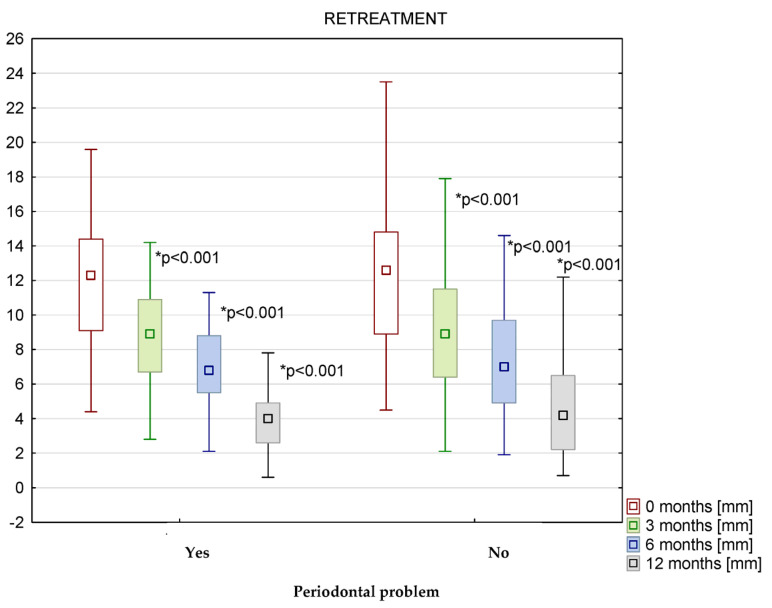
The thickness of changes in Schneiderian membrane in mm from 0 to 12 months after secondary treatment with and without periodontal problems.

**Table 1 medicina-58-01302-t001:** Characteristics of the study participants (n = 474).

		ALL (N = 474)	MALES (N = 280)	FEMALES (N = 194)
		N (%)		
AGE (YEARS)	18-35	146 (30.8)	62 (32.0)	84 (30.0)
	36-50	225 (47.5)	91 (47.0)	134 (48.0)
	51-65	103 (21.7)	41 (21.0)	62 (22.0)
PRIMARY ROOT CANAL TREATMENT (N = 289)	Type of tooth			
	Premolar	130 (44.9)	62 (48.8)	68 (41.9)
	Molar	159 (55.1)	65 (51.2)	162 (58.0)
	Periodontal problem			
	Yes	173 (59.9)	74 (58.3)	99 (61.1)
	No	116 (40.1)	53 (41.7)	63 (38.9)
ENDODONTIC RETREATMENT (N = 185)	Type of tooth			
	Premolar	96 (51.9)	29 (43.3)	67 (56.8)
	Molar	89 (48.1)	67 (56.7)	118 (43.2)
	Periodontal problem			
	Yes	89 (48.1)	35 (52.2)	54 (45.8)
	No	96 (51.9)	32 (47.8)	64 (54.2)

## Data Availability

Non-digital data supporting this study are curated by Klaudia Migas.

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
