# Peer review of "Healing of Unilateral Maxillary Sinusitis by Endodontic and Periodontal Treatment of Maxillary Teeth"

_medicina, 2022, doi:10.3390/medicina58091302_

Round 1

Reviewer 1 Report

Comment from reviewer

Title

-       Too general, please be more specific to the objective or content of the study.

Abstract

-       Please add the method for determination of the inflammation into the abstract since it is a key parameter of this study.

Introduction

-       Please review the current state of this research field and cite the key publications in the introduction part.

-       The prevalence of tooth-borne sinusitis should be reviewed, both pulp canal infection and periodontal infection. 

Materials & Methods

-       Ethical consideration: In case of non-interventional studies, the author guidelines indicate that If ethical approval is not required, authors must either provide an exemption from the ethics committee or are encouraged to cite the local or national legislation that indicates ethics approval is not required for this type of study.

-       Was there any sample size calculation? Please identify in the context.

-       For inclusion criteria, why was the multiple-visit RCT not included in the study? 

-       The manufacturer, city, country of all equipment and programs used in the study must be shown in the context.

-       Was there any inter-examiner calibration in the study? Please identify in the context and results.

-       For CBCT analysis, please describe more details about the measurement of the thickness of Schneiderian membrane. Which plane did you choose? How did you identify and measure the highest value of the thickness? Please add one figure that shows an example of the measurement.

Results

-       Figures 2 and 3 should be removed since there is no significant difference between male and female.

Discussion

-       From the results, the average thickness of Schneiderian membrane was higher than 2 mm. Does it mean the inflammation is not fully diminished?

Conclusion

-       The conclusion is too long. It should be concise and contain only the main conclusions regarding the results of the research.

Reference

-       All references must be in a consistent format.

Language 

-       Please check the punctuations.

-       Please check the missed placed comma.

-       Please check the spelling, especially the words “premoral” and “moral”.

-       Please use full stop only after the full sentence.

Reviewer 2 Report

Reviewer’s comment

This is a highly relevant research to any dentists concerning about maxillary sinusitis in association to dental causes. The suggestions given to the time for review after treatment are also clinically applicable. Some comments to improve or clarify this research paper as follows:

Introduction

·      The following sentences or statement will require rephrasing to avoid confusion:

·      Lines 80: “…. with possible presence of the possible serous and purulent discharge.”

·      Lines 93-94: “Confirmation of the diagnosis, which was established during the anamnesis and physical examination, is provided by radiological diagnostics.”

·      Lines 109-115: “The aim of this study was to determine time from the ……..The relationship between variables like a type of dental treatment, including primary root canal treatment vs retreatment and periodontal treatment was investigated.”

·      Lines 50-51: Do you actually mean “….increasing size”?

·      Lines 62-63: Another school of thought is that the periodontal disease has to involve the apical foramen. Hence, worthwhile mentioning and add reference e.g. Langeland K, Rodrigues H, Dowden W (1974) Periodontal disease, bacteria, and pulpal histopathology. Oral Surgery, Oral Medicine, Oral Pathology 37, 257-270.

·      Line 64:  “3D cone-beam computerized tomography”?

Materials and methods

·      Was there any calibration exercise before actual CBCT assessment?

·      Was the assessment performed in a blinded manner?

·      Lines 138-139: “Drug use…”, “Disease…” - perhaps need more elaboration of as in drug use and diseases for any systemic conditions or just maxillary sinusitis?

·      Lines 150 & 169: “Carestream 9300 Premium C device” and “Statistica 12 (StatSoft) and StatXact (Cytel) software” - need to specify manufacturer, city, country.

·      Line 163: “All measurements were made according to a standardized method” - reference needed.

Results

·      Repeated typos for “premolar”, “molar” and “periodontal problem”.

·      May consider putting asterisk * in the Figures' (charts) to demonstrate the group/s with significance difference; and also indicate between which groups such significant difference exists.

·      Lines 261 & 290: What is referred to by "primary treatment?" Is it primary root canal or periodontal treatment? Please specify.

·      Line 194: Please specify the “tooth mobility” classification used.

·      The following sentences or statements will require rephrasing to avoid confusion:

·      Lines 184-186: “Based on the medical records, ……..there were no such a disproportions”.

·      Lines 206-208: “The greatest reduction of inflammation ……after 12 months of Schneiderian membrane”.

·      Lines 220-222: “At baseline the Schneiderian membrane was ……. The same differences were observed both, in males and females with p<0.001 (d=1.98 and d=1.88, respectively)”.

·      Lines 239-243: “In patients after primary treatment, the greatest reduction of inflammation …… In patients with periodontal problem…….”

·      Lines 288-289: no hyphens for “periodontal” and “membrane”.

Discussions

·      There seems to be some repetitions of the same messages throughout the discussion.

·      Line 352: “According to the guidelines, the recommended x-ray follow-up…..” – please add reference.

Conclusions

·      Should be more concise and precise, hence it might be better to cut back to one short paragraph with the most important concluding remarks.

Round 2

Reviewer 1 Report

This manuscript is acceptable but may need some minor revision.

Reviewer 2 Report

There has been a marked improvement. Please see additional comments as follows:

Line 20: Cone Beam (not Bean)

Line 98-99: radiologic images (instead of pictures)

P118-119: hypothesis statement remains confusing

Line 156-159: the entire paragraph needs to be rephrased for clarity

P160-163: the entire paragraph needs to be rephrased for clarity

Line 182-186: 'axial' and 'cross-sectional' are synonym

Line 209: "who assessing CBCT" grammatical error

Line 210: "Editing was conducted ...." - what do you mean by editing?

Line 258: "..of Schneiderian membrane" should be deleted

Line 332: "...with" should be deleted

Figure 3 and 4: not clearly labelled, i.e. what do the two groups belong to? Are they with and without periodontal problems?
